# Targeted Therapies and Drug Resistance in Advanced Breast Cancer, Alternative Strategies and the Way beyond

**DOI:** 10.3390/cancers16020466

**Published:** 2024-01-22

**Authors:** Andrea Nicolini, Paola Ferrari

**Affiliations:** 1Department of Oncology, Transplantations and New Technologies in Medicine, University of Pisa, 56126 Pisa, Italy; 2Unit of Oncology, Department of Medical and Oncological Area, Azienda Ospedaliera—Universitaria Pisana, 56125 Pisa, Italy; p.ferrari@ao-pisa.toscana.it

**Keywords:** advanced breast cancer, targeted therapies, alternative therapies

## Abstract

**Simple Summary:**

“Precision medicine” is a therapeutic strategy launched over two decades ago. It relies on drugs that inhibit key molecular mechanisms/pathways or induce anti-tumour effects by modulating immune suppression. During this time, the complexity of advanced breast cancer disease has been increasingly elucidated and many clinical trials, sponsored by multinational drug companies, have been carried out. Nevertheless, patients have seen limited benefits from these clinical trials and the few approved drugs are costly. Concomitant, although other therapeutic strategies have been proposed by researchers over time, the resources available for alternative research have been restrained. As to this, while drug repurposing is an obvious answer to expensive targeted therapies, counteracting micro-metastatic disease represents a new field and seems to be the “way beyond” the current research.

**Abstract:**

“Targeted therapy” or “precision medicine” is a therapeutic strategy launched over two decades ago. It relies on drugs that inhibit key molecular mechanisms/pathways or genetic/epigenetic alterations that promote different cancer hallmarks. Many clinical trials, sponsored by multinational drug companies, have been carried out. During this time, research has increasingly uncovered the complexity of advanced breast cancer disease. Despite high expectations, patients have seen limited benefits from these clinical trials. Commonly, only a minority of trials are successful, and the few approved drugs are costly. The spread of this expensive therapeutic strategy has constrained the resources available for alternative research. Meanwhile, due to the high cost/benefit ratio, other therapeutic strategies have been proposed by researchers over time, though they are often not pursued due to a focus on precision medicine. Notable among these are drug repurposing and counteracting micrometastatic disease. The former provides an obvious answer to expensive targeted therapies, while the latter represents a new field to which efforts have recently been devoted, offering a “way beyond” the current research.

## 1. Introduction

A commonly accepted molecular classification of primary breast cancer includes subtypes such as ER+/HER2− luminal, HER2+, and triple-negative breast cancer (TNBC). Among these, ER+/HER2− luminal breast cancer is divided into luminal A and luminal B subtypes based on the KI-67 nuclear molecular marker, according to whether this nuclear molecular marker is <25% or >25%, respectively [1]. ER+/HER2− luminal breast cancer accounts for 60% to 80% of all breast malignancies, with its incidence increasing with age [2,3]. Currently, the first-line treatment for ER+/HER2− luminal and HER2+ breast cancer patients respectively combines antiestrogens with cyclin-dependent 4/6 kinase (CDK 4/6) inhibitors and monoclonal antibodies against specific HER2 epitopes, along with chemotherapy. For the more aggressive TNBC subtype, the standard first-line therapy includes chemotherapy with platinum-derived compounds, with or without PD1/PD-L1 inhibitors [4]. The introduction of “targeted therapies” or “precision medicine” in 2001, first reported by Slamon et al. in metastatic breast cancer patients overexpressing HER2, marked a new era [5]. In this study, a monoclonal antibody against HER2 combined with conventional chemotherapy was used, thus introducing the era of “targeted therapies” (also called “precision medicine”). Since then, this novel therapeutic strategy has proliferated [6,7], accompanied by a growing number of studies focused on molecular signaling that promotes the main cancer hallmarks. More than two decades later, this paper examines some principal issues with this therapeutic strategy in advanced breast cancer and provides insights into the way forward.

## 2. Precision Medicine and Drug Resistance in Breast Cancer

“Precision medicine”, heavily supported by multinational drug companies, has revolutionized breast cancer treatment. This approach, promising the right drug for the right disease, has significantly enhanced our understanding of cancer and provided more effective treatment options in selected subpopulations. However, it has also constrained oncologic research due to its focus and resource allocation, and a large part of the scientific community was involved in the new strategy. So, many studies were devoted to translating a better comprehension of cancer obtained through basic research into clinical practice [8].

The treatment of metastatic breast cancer, which develops in about 20% to 30% of breast cancer patients and is generally incurable, has been particularly impacted. While precision medicine has improved cancer knowledge and introduced new treatment avenues, it has also led to significant research limitations.

### 2.1. The Most Common Targeted Therapies

The core principle of precision medicine is to test drugs in randomized clinical trials that target specific molecules, molecular pathways, or genetic/epigenetic alterations crucial for promoting cancer hallmarks [8]. These targeted therapies either directly inhibit tumor-promoting mechanisms [9] or induce antitumor effects by modulating immune suppression [10]. The PI3K-AKT-mTOR pathway inhibitors in TNBC, CDK 4/6 inhibitors in HR+ HER2− breast cancer, and monoclonal antibodies against HER-2 alone or associated with chemotherapy or a drug in HER2+ breast cancer are examples of the former, and PARP inhibitors again in TNBC are an example of the latter modality. The most commonly targeted pathways and their corresponding drugs are illustrated in Figure 1 and Figure 2. Advanced disease often serves as the initial setting for therapeutic investigation, and upon successful results, the drug is tested in the neoadjuvant/adjuvant setting.

#### 2.1.1. PI3K-AKT-mTOR Inhibitors

The involvement of altered PI3K/AKT/mTORC1 signaling in endocrine resistance suggested the combination of PI3K/AKT/mTORC1 inhibitors with conventional endocrine therapy [11,12]. The Bolero-2 trial investigated the efficacy and safety of the mTORC1 inhibitor everolimus in advanced HR+ HER2− breast cancer patients after relapse or progression on nonsteroidal aromatase inhibitors (NSAIs) [13,14]. Alpelisib, a PI3Kalpha-specific inhibitor, when combined with fulvestrant in the SOLAR-1 trial, significantly improved progression-free survival compared to fulvestrant alone in the subset with the PIK3CA mutation [15]. A recent review comparing the two drugs concluded that “adding either compound to standard endocrine therapy provided similar PFS advantage”. In the SOLAR-1 trial, grade (G) 3 or 4 (G3/G4) adverse events (AEs) occurred in 76% of patients, while they were observed in 42% of patients in the BOLERO-2 trial [16]. The PI3K-AKT-mTOR pathway is often hyperactivated in TNBC, mainly due to the downregulation or absence of the tumor suppressor phosphatase and tensin homolog (PTEN), commonly accompanied by a poor outcome [17,18,19]. Activating PIK3CA mutations and PTEN aberrations are found in approximately 10% and 30–50% of TNBC patients, respectively [17]. Also, PIK3CA and AKT1 mutations are reported to occur more often in AR-positive TNBC [17]. In aggressive TNBCs with PTEN deficiency, the combination of PI3K-AKT-mTOR pathway inhibitors with conventional chemotherapy is considered a helpful treatment option. Everolimus has shown efficacy against TNBC in preclinical studies, and several phase I-II clinical trials evaluating mTOR and PI3KA inhibitors in advanced TNBC are ongoing (NCT02531932, NCT01931163, NCT01629615, NCT04216472). Additionally, AKT has been identified as a relevant therapeutic target in advanced/metastatic TNBC patients, with the AKT inhibitor ipatasertib, when combined with paclitaxel, improving PFS and overall survival (OS) compared to paclitaxel alone [20,21]. Uprosertib and capivasertib are further AKT inhibitors; the former is undergoing evaluation in a phase II clinical trial carried out in metastatic TNBC (NCT01964924), while the latter, given in a different phase II trial in association with chemotherapy, prolonged the OS of the same population of patients [22].

#### 2.1.2. Cyclin Kinase Inhibitors 

Recent recognition of the critical role of the CCND1-CDK4/6-RB pathway in regulating the G1–S phase transition has led to the proposal and subsequent validation of CDK 4/6 inhibitors as significant in improving clinical outcomes for ER+ HER2− metastatic breast cancer patients on first-line hormonal therapy [23,24]. As first-line salvage treatment, these inhibitors, in combination with aromatase inhibitors (AIs) or fulvestrant, have demonstrated efficacy and are recommended in this patient population. Clinical trials have shown a median PFS extending from 25.3 months with ribociclib [25,26] to 28.2 months with abemaciclib [27]. Although median OS remains unreported in the abemaciclib trial, a significant difference was found in more recent trials between treated patients and controls for ribociclib (63.9 vs. 51.4 months, *p* = 0.008) [25,26] and not for palbociclib (53.9 vs. 51.2 months) [28,29]. It is noteworthy that more than 10% of patients receiving CDK4/6 inhibitors experienced grade 3–4 adverse events. Furthermore, abemaciclib is currently being evaluated in the adjuvant setting for the high-risk HR+ HER2− breast cancer subset. Table 1 summarizes the results of major clinical trials conducted with PI3K-AKT-mTOR pathway and CDK 4/6 inhibitors in advanced breast cancer.

#### 2.1.3. Monoclonal Antibodies (mAbs) against HER-2 Combined with Conventional Chemotherapy (CT) and Antibody–Drug Conjugates (ADCs)

The HER2+ subtype of breast cancer, known for its poorer prognosis compared to luminal breast cancer [30,31], saw a significant treatment advancement in 1998 with the approval of trastuzumab for this subpopulation of metastatic breast cancer [5,32]. This monoclonal antibody (mAb), when combined with first-line chemotherapy, inhibits HER2 signaling and the PI3K pathway, leading to G1 cell-cycle arrest, apoptosis, and antiangiogenic effects. Subsequent developments introduced antibodies targeting different HER2 epitopes and engaging with Fc receptors [33] capable of inducing antibody-dependent cellular cytotoxicity (ADCC). Pertuzumab, the most well known of this new generation of anti-HER2 mAbs [34,35] significantly improved the overall response rate (ORR) and, for a few months, the PFS and OS in HER2+ metastatic breast cancer [36,37] in combination with trastuzumab and docetaxel. In another study that recruited locally recurrent or metastatic HER2+ breast cancer patients, the effectiveness and safety of pertuzumab and trastuzumab joined with a taxane (docetaxel, paclitaxel, or nab-paclitaxel) at investigator choice were the main endpoints. In this trial, similar results were reported irrespective of the taxane used [38,39].

Despite these advances, constitutive (intrinsic or “de novo”) or acquired drug resistance as a result of initial genetic aberrations or arising during disease progression, respectively, limits the effectiveness of anti-HER2 antibodies, benefiting only about 30% of HER2+ tumors [40]. Recent focus has shifted towards antibody–drug conjugates (ADCs) like trastuzumab emtansine (T-DM1) [41,42] and more recently trastuzumab deruxtecan (T-DXd) [43] and sacituzumab govitecan [44], a new category where a tumor-specific mAb covalently conjugates a cytotoxin involving microtubules, thus enhancing immunity against tumors. T-DXd and T-DM1, built up as a further strategy against anti-HER2 antibody resistance, are commonly recommended for treating patients upon progression with pertuzumab and/or trastuzumab and a taxane. In particular, T-DM1 comprises emtansine, a microtubule inhibitor, bound to trastuzumab via a nonreducible thioether link [45]. In a pivotal clinical trial, T-DM1 was compared with lapatinib given in addition to capacitabine. T-DM1 significantly increased median PFS (9.6 vs. 6.4 months) and median OS (30.9 vs. 25.1 months) compared with the combination [46]. In another clinical trial, T-DM1, opposed to physicians’ choice, again significantly prolonged median PFS (6.2 vs. 3.3 months) and median OS (22.7 vs. 15.8 months) [47,48]. The efficacy of T-DXd, which comprises an anti-HER2 antibody, a cleavable tetrapeptide-based linker, and a cytotoxic topoisomerase I inhibitor, was assessed in two trials. In one of them, a phase II study where HER2+ metastatic breast cancer patients had formerly received T-DM1, T-DXd showed prolonged antitumor activity [38]. In the other, a phase III, multicenter, open-label randomized trial, the safety and effectiveness of T-DXd were compared with those of T-DM1 in subjects who were before given trastuzumab and a taxane. The authors concluded that those who were given T-DXd were more likely to have a lower risk of disease progression or death than those who were given TD-M1 [49]. Additionally, and importantly, T-DXd also showed effectiveness in HER2-low breast cancer patients, and in an early clinical trial by Modi et al. [43], an ORR of 37% in this subset was reported. Similarly, in the DAISY trial [50], T-DXd was investigated in metastatic breast cancer with different HER2 levels. In this study, the ORRs were 70.6%, 37.5%, and 29.77% in overexpressing (3+), HER2-low (2+), and HER2-nonexpressing patients, respectively. Therefore, novel agents in patients with HER2-low tumors are under investigation for potential clinical benefit (NCT04400695, NCT04742153, NCT05831878, NCT05904964).

#### 2.1.4. Antibody–Drug Conjugates (ADCs), ICIs, and PARPis in TNBC

Trophoblast antigen 2 (Trop-2) is a growth-signaling-promoting glycoprotein that is hyperexpressed in many epithelial cancers [51]. Sacituzumab govitecan-hziy, an anti-Trop-2 antibody linked with an active metabolite of irinotecan (SN-38) [44], has been shown to increase objective response rates (ORRs) and median PFS in heavily pretreated HR+/HER2− or metastatic triple-negative breast cancer (TNBC) patients, compared to conventional chemotherapy (33.3% and 5.5 months vs. 10–15% and 2–3 months, respectively) [44,52,53,54]. Specifically, in the phase III ASCENT randomized trial (NCT02574455), metastatic TNBC patients treated with this antibody–drug conjugate (ADC) experienced a median PFS of 5.6 months (95% CI, 4.3–6.3), versus 1.7 months for those given chemotherapy as selected by the investigator (*p* < 0.001) [55]. In 2020, the FDA granted accelerated approval to sacituzumab govitecan-hziy for advanced/metastatic TNBC following heavy pretreatment.

Immune checkpoint inhibitors (ICIs) target immune checkpoints, key mechanisms that maintain immune homeostasis [56,57], and account for checkpoint blockade (ICB). Signals such as cluster of differentiation 28 (CD28) and cytotoxic T lymphocyte associated antigen 4 (CTLA-4) play a pivotal role in regulating T-cell activity; CTLA-4 signaling, in contrast to CD28, which is necessary to activate T cells and for cytokine release, inhibits T-cell activation. PD-1, CTLA-4, and CD28 are receptors on effector T cells and bind with their ligands, namely CD274 (PD-L1) and CD273 (PD-L2) for PD-1 and B7-1 (CD80) or B7-2 (CD86) among others for CTLA-4 and CD28 [58,59]. PD-L1 and CTLA-4 upregulation was described in HER2+ and TNBC [60,61]. Therefore, ICB by ICIs has emerged as a novel strategy to lift the “brake” on the antitumor immune response. Clinical studies, primarily in advanced or metastatic TNBC patients and to a lesser extent in HER2+ or other molecular subtypes, have investigated ICIs such as PD-1 (pembrolizumab, nivolumab), PD-L1 (atezolizumab, durvalumab, avelumab), and CTLA-4 (tremelimumab, ipilimumab) inhibitors or are ongoing.

Approximately 10–20% of TNBC patients exhibit a BRCA1/2 germline (gBRCA1/2) mutation, and PARP inhibitors (PARPis) are recommended for these individuals. BRCA1/2 genes encode tumor suppressor proteins that facilitate DNA repair through homologous recombination; mutations in these genes lead to homologous recombination deficiency (HRD). A subset of patients without BRCA1/2 mutations, termed “BRCAness”, may also exhibit HRD due to epigenetic BRCA inactivation or somatic mutations in other key genes. Clinically, patients with the BRCAness phenotype share similarities with those who have BRCA mutations [62].

Table 2 [63,64,65,66,67,68,69,70,71] summarizes the main characteristics and findings from principal clinical trials that have investigated anti-HER2 monoclonal antibodies (mABs), ICIs, and PARPis as first-line therapies in advanced breast cancer.

### 2.2. Main Mechanisms of Drug Resistance

Drug resistance, whether intrinsic or acquired, is a primary reason for metastatic cancer’s incurability. Research has focused on understanding the mechanisms responsible for this resistance, particularly in aggressive triple-negative breast cancer (TNBC), though resistance in HER2+ and ER+/HER2− subtypes has also been extensively investigated.

#### 2.2.1. TNBC

We have recently and extensively reported on the main mechanisms of drug resistance in TNBC [9]. The induction of cancer stem cells (CSCs) following neoadjuvant chemotherapy (NACT), the presence of ATP-binding cassette (ABC) transporters, hypoxia, escape from apoptosis, tyrosine kinase receptors, a disintegrin and metalloproteinase 10 (ADAM10), noncoding RNAs (ncRNAs), DNA methylation, and phosphoproteome alterations, including kinase phosphorylation, are among these mechanisms.

The quiescent state may be the principal reason why CSCs are refractory to cytotoxic agents, which are usually more effective against proliferating cells. Moreover, CSCs overexpress ABC transporters, which confer resistance to many cytotoxic drugs. Multidrug resistance proteins (ABCC1/MRP1), breast cancer resistance protein-2 (ABCG2/BCRP), and multidrug resistance protein-8 (ABCC11/MRP8) are upregulated in TNBC, increasing the efflux of most administered chemotherapeutics. Hypoxia and acidity in the tumor microenvironment (TME) promote the CSC phenotype, thereby fostering chemoresistance. Evasion from apoptosis through aberrations of the prosurvival MCL-1 gene often occurs in residual TNBCs following chemotherapy, and a relationship between Mcl-1 expression and chemoresistance has been suggested.

The epidermal growth factor receptor (EGFR) and insulin growth factor-1 receptor (IGF-1R) are part of the tyrosine kinase family and regulate the PI3K-AKT-mTOR and Janus kinase (JAK)/signal transducer and activator of transcription (STAT) pathways involved in TNBC chemoresistance. Epigenetic remodeling through the altered expression of microRNAs (miRNAs) and long noncoding RNAs (lncRNAs), as well as DNA methylation, can induce chemotherapy resistance in TNBC. Fundamentally, the amount of phosphoproteins is a crucial cellular feature for signal transduction. In an experimental investigation conducted on two cell lines sensitive to taxanes, anthracyclines, gemcitabine, and cisplatin and four resistant cell lines, differentially phosphorylated cyclin-dependent kinases collaborated to induce epithelial–mesenchymal transition (EMT) in the cell lines resistant to the cytotoxic agents. For more details, see elsewhere [9].

#### 2.2.2. HER2+ Breast Cancer

A variety of mechanisms of resistance to HER2-targeted therapies have been identified. Reduced HER2 expression and/or mutations can impair mAb binding, diminishing therapeutic efficacy [72]. Structural variants, such as the p95HER2 isoform produced by ADAM10 cleavage, prevent trastuzumab binding. Intracellular HER2 domain mutations, like L755S, confer resistance to lapatinib and other therapies. L755S, V777L, D769Y, and K753E represent mechanisms of resistance to trastuzumab [72]. Constitutive activation mutations within the PI3K-AKT-mTOR pathway allow cells to bypass the blockade of signaling by anti-HER2 treatments [72]. Mutations in the MAP kinase pathway can shift cellular dependence from the PI3K/AKT to the MEK/ERK pathway, leading to anti-HER2 therapy resistance [73]. Additionally, increased activation of parallel pathways such as ER hyperactivity boosts the PI3K-AKT-mTOR pathway and can result in acquired resistance [74]. Altered SRC activity and an overactive cyclin-dependent kinase 4/6 (CDK 4/6) axis further contribute to resistance [75]. RTK heterodimerization with HER2, particularly HER2/EGFR and HER2/HER3, facilitated by neuroregulin-1 (NRG1), can circumvent HER2 homodimerization inhibition, the target of anti-HER2 mAbs. Accordingly, NRG1 has been reported to join with resistance to T-DM1 and to TKIs [76]. ER+ signaling has been reported to engage in crosstalk with the HER2 pathway, and this crosstalk is significant, as around 50% of HER2+ breast cancers also overexpress ER+, influencing therapeutic response and outcomes. Further, the upregulation of different proteins with regard to the balance between cell death and survival could replace the impaired HER2 function [76].

Moreover, drug efflux pumps, such as ABCG2/BCRP and MRP, mainly reduce the cytotoxicity of ADC warheads like DXd and DM. Variations in HER2 expression within tumor cells can also impair the activity of anti-HER2 drugs [76,77]. A key aspect of trastuzumab’s effectiveness is triggering ADCC when its Fcγ region is recognized by immune cells expressing FcγRs. However, polymorphisms in the Fcγ receptor may counteract trastuzumab’s immune activity, while the amount of immune cells in the TME can regulate ADCC [78]. Margetuximab, a human/mouse chimeric IgG1 anti-HER2 mAb, is engineered to enhance binding to the activating receptor FcγRIIIA (CD16A) and reduce interaction with the inhibitory Fc receptor FcγRIIB (CD32B), thus maximizing ADCC activity. This drug is composed of trastuzumab with an additional engineered Fc-domain, where five amino acids are replaced [79]. While not all of the reasons for resistance to anti-HER2 antibody therapies are fully understood, the constitutive activation of downstream pathways is believed to play a critical role [76]. Therefore, combining anti-HER2 therapies with more conventional treatments is often recommended in clinical practice to circumvent some of these resistance mechanisms.

#### 2.2.3. CDK 4/6 Inhibitors in ER+ HER2− Molecular Subtype

Studies suggest that alterations in cell cycle regulatory molecules and cancer cells’ shift to alternative pathways are common mechanisms of resistance to CDK 4/6 inhibitors. Hyperexpression of CDK6/7, upregulation of cyclin E, loss of retinoblastoma (Rb) function, and dysregulation of the PI3K-AKT-mTOR signaling pathway are frequently implicated. Estrogen facilitates the G1 to S phase transition by overexpressing cell cycle modulators such as Cyclin D1 and Cyclin E or activating CDK2/4, which in turn activates Rb—the downstream substrate [80]. Endocrine treatments arrest the cell cycle at the G1 phase by affecting the transcription of ER-dependent cell cycle regulators [80,81]. In tumors resistant to endocrine treatment, the activation of c-Myc, Cyclin D1, or CDK4/6 often occurs [82], allowing the bypass of antiestrogen inhibition of the G1 to S transition. The inhibition of p21 transcription by antiestrogens, leading to activation of Cyclin E/CDK2, has been identified as a primary mechanism of c-Myc-induced endocrine resistance [83]. CCND1 activation of CDK4/6 kinases and its upregulation is associated with ER positivity, poor outcomes, and resistance to endocrine therapy in ER+ breast cancer patients [84]. While preclinical studies demonstrated the efficacy of CDK4/6 inhibition in Rb-proficient human tumor xenograft models, the introduction of CDK4/6 inhibitors has significantly changed the management of endocrine-resistant ER+ breast cancer [85]. CDK4/6 inhibitors prevent Rb hyperphosphorylation, thus arresting the cell cycle at the G1 phase [86]. As G1 to S progression is crucial for acquiring endocrine resistance, CDK4/6 inhibitors were expected to significantly enhance the effectiveness of antiestrogens [86,87,88]. Proteins that govern CDK4 or Rb have also been found to regulate resistance. For example, ankyrin repeat and LEM domain-containing 2 (LEM4), a nuclear envelope protein, promotes CDK4 and Rb as well as Aurora A-mediated ER phosphorylation, enhancing ER-dependent Cyclin D1 and c-Myc expression, leading to tamoxifen resistance [89]. The efficacy of CDK4/6 blockade is likely increased by inhibiting receptor tyrosine kinases, due to their interaction with pathways that promote proliferation. A simultaneous FGFR1/CCND1 amplification and synergistic inhibition of tumor growth occur in ER+ breast cancer cells deprived of estrogen when palbociclib is combined with an FGFR inhibitor [90]. In line with this, clinical trials investigating the combination of endocrine therapy with CDK4/6 and PI3K/mTOR inhibitors have reported tolerable side effects and promising clinical outcomes [91,92].

### 2.3. Prognostic and Predictive Biomarkers

While ER, PR, HER-2, and MIB1/Ki67 remain significant prognostic biomarkers in breast cancer, advanced technologies like wide next-generation sequencing (WNGS) have enabled the discovery of additional prognostic genes and signatures. In a study conducted in early breast cancer and normal tissues where gene expression was examined, 16 differentially expressed, including 2 upregulated and 14 downregulated genes, were significantly associated with prognosis [93]. In another study, meta-analysis and bioinformatics analyses found that secreted protein acidic and rich in cysteine (SPARC) expression was a good prognostic indicator [94]. Also, the prognostic implication of ferroptosis-related genes like ACLS4 and GPX4 for breast cancer treated with neoadjuvant chemotherapy (NACT) has been reported [95]. Bioinformatics analyses have also highlighted the prognostic value of the paired like homodomain transcription factor 1 (PITX1) [96], a macrophage marker gene signature [97], and lncRNA signatures associated with immune infiltration and tumor mutation burden (TMB) [98]. Moreover, metabolic signatures related to energy metabolism have been used to categorize breast tumors into distinct prognostic clusters. In particular, energy-related metabolic signatures allowed for distinguishing breast tumors into cluster 1 with elevated glycolysis and a lower survival rate and cluster 2 with increased fatty acid oxidation and glutaminolysis [99]. Predictive biomarkers are crucial in precision medicine, guiding the use of expensive drugs that are ineffective for most unselected patients. PI3K-AKT-mTOR inhibitors, for example, show more benefit in patients with PTEN-low or pathway-altered tumors [21]. CDK 4/6 inhibitors are particularly effective in the HR+ HER2− breast cancer subset. HER-2 mAbs, combined with chemotherapy or ADC, are preferred treatments for advanced breast cancer with HER2 expression/amplification, and PD-L1 inhibitors coupled with chemotherapy are suitable for advanced TNBC with PD-L1 expression. Multiple PD-L1 IHC assays with different scoring algorithms exist, necessitating companion diagnostic tests for drug approval and patient selection [100]. Patients with advanced TNBC harboring BRCA1/2 mutations or the BRCAness phenotype are targeted with platinum-derived compounds and PARP inhibitors. Intensive investigations to define predictive biomarkers for other potential targeted therapies are ongoing [9]. Despite these advances, not all patients benefit from targeted therapies, and efficacy varies among responders.

### 2.4. The Main Limitations

Targeted therapy strategies face major challenges, including the occurrence of adverse events (AEs) and both constitutive (intrinsic or “de novo”) and acquired drug resistance. These therapies, often administered alongside standard chemo and endocrine treatments, can significantly reduce patients’ quality of life (QoL) due to increased AEs. In many cases, severe AEs lead to a temporary or permanent discontinuation of treatment. For instance, patients treated with everolimus and exemestane experienced higher AE rates, dose modifications, and treatment discontinuations compared to those receiving a placebo. The mortality rate due to AEs in the everolimus group was 1.4%, versus 0.4% in the placebo group [13]. Similarly, Slamon et al. [5] reported that a notable proportion of patients experienced cardiac dysfunction, with varying incidences depending on the combination of treatments received. Namely, 63 patients showed symptomatic or asymptomatic cardiac dysfunction; 39 of 143 patients (27%) had received an anthracycline, cyclophosphamide, and trastuzumab; 11 (8%) of 135 had received an anthracycline and cyclophosphamide alone; 12 (13%) of 91 had received paclitaxel and trastuzumab; and 1 (1%) of 95 had received paclitaxel alone. Despite targeted therapies often being palliative, they can worsen QoL, with only modest extensions in PFS and, at best, a one- to two-year increase in OS, as outlined in Table 1 and Table 2.

## 3. Other Than Targeted Therapies

Given the high cost/benefit ratio of targeted therapies, alternative therapeutic strategies have gained attention. Notable among these is the focus on breast cancer stem cells (CSCs), T-lymphocyte infusion, exosomes as drug nanocarriers, and cancer cell reprogramming. All of them use substantially different research approaches that rely upon specific cells such as CSCs and T lymphocytes or cellular products such as exosomes or stem cell differentiation factors. CSCs are central to chemoresistance [9] due to both intrinsic factors, like oncogenic pathway activation, and extrinsic factors, such as vascular niches, hypoxia, stromal cells, and the extracellular matrix. T-lymphocyte infusions, enhanced through genetic engineering (modified T-cell receptors or chimeric antigen receptors) to target specific tumor epitopes [10], also show promise. Exosomes, as nanocarriers, facilitate intercellular communication, while cancer cell reprogramming leverages differentiation factors to prevent cancer progression [101]. Additionally, drug repurposing emerges as a cost-effective alternative to current therapies, and combating micrometastatic disease represents a new frontier in research.

### 3.1. Drug Repurposing, an Easier and Cheaper Alternative Strategy

Drug repurposing, also known as repositioning, seeks new uses for existing drugs, whether marketed, discontinued, or previously shelved [102,103]. This strategy offers a cost-effective and expedited route for drug development by utilizing FDA-approved or investigational drugs for alternative indications [102,103]. Initially aimed at treating rare genetic diseases, drug repurposing has evolved to address a broader spectrum of conditions, including cancer. This shift reflects a growing understanding that drugs may interact with multiple molecular targets, influencing various biological processes and diseases. Conversely, for a long time, “one drug, one target” has been the paradigm of drug discovery. The behavior of any system and its biological processes depend on protein–protein interactions, proteins interacting with other biomolecules, and finally the complex network of interactions that at the same time allows the expansion of the number of potential targets [104]. So, unlike the initial paradigm, the current drug repurposing is based on the knowledge that a drug can address different targets, different diseases can have molecular similarities, and a target can display multiple effects with regard to molecular function [102]. Usually, the development of novel drugs/agents is expensive and time-consuming; therefore, this is a major challenge and accounts for the high cost of the few drugs entering clinical practice. Thus, while drug repurposing began by chance to cure rare genetic diseases, currently, it is the most reasonable alternative approach to targeted therapies against human malignancies. In fact, it allows the development of further therapeutic choices for cheaper cancer patient treatment in clinics. Regarding this, it must be considered that in most underdeveloped countries around the globe, expensive anticancer drugs do not allow the medical needs of cancer patients to be fully met. The rapid increase in the number of reports on drug repurposing as well as the number of dedicated journals is the most convincing proof of the increasing interest in this emerging strategy [102]. Metformin, a biguanide given for type 2 diabetes mellitus, and aspirin, commonly used as an anti-inflammatory drug, are two examples of drugs that have been repurposed for their anticancer properties. So far, antidiabetics, antibiotics, antifungal drugs, anti-inflammatory drugs, antipsychotic drugs, PDE inhibitors, estrogen receptor (ER) antagonists, antiparasitic drugs, Antabuse, and cardiovascular agents/drugs have been the categories of drugs more commonly involved in repurposing procedures due to their anticancer capabilities [103]. The substantial support from nonprofit organizations (NPOs) and the development at various phases of advance [104] of approximately 170 repurposed drugs from 2012 to 2017 [105] signify the strategy’s growing prominence and potential to meet the oncologic needs of patients, especially in underdeveloped countries where costly cancer treatments are less accessible. The use of computational techniques is further accelerating the discovery of new drug applications, contributing to the strategy’s appeal and expansion.

### 3.2. The Way Beyond: Micrometastatic Disease as the Principal Target

While “targeted therapies” have traditionally focused on addressing clinically overt metastatic or recurrent disease, our new therapeutic strategy aims to preserve the health of high-risk and locally advanced breast cancer patients by preventing the progression of micrometastases to a clinically or radiologically detectable state. This approach aligns with the mathematical model initially proposed by Goldie and Coldman in 1979. Consistent with this, traditionally, the first four to six months after primary radical surgery have been considered ideal for adjuvant chemotherapy. However, over the last two decades, advances in genetics and molecular biology have significantly altered our understanding of cancer. It is no longer viewed as solely a loco-regional disease with distant metastasis occurring after primary tumor removal. Instead, extensive evidence indicates that cancer cells often spread to distant organs at an in situ stage well before the primary diagnosis of a tumor [106]. When cancer cells metastasize to distant organs through circulation, they interact with various microenvironments, contributing to the formation of premetastatic niches (PMNs). This process involves three successive phases: (a) education of the metastatic microenvironment by the primary tumor, (b) recruitment of immune-inhibiting cells to metastatic sites, and (c) migration of tumor cells from circulation to PMNs [107,108]. This entire process, known as cancer cell “homing”, is regulated by gene expression and the secretion of chemokines. Additionally, extracellular vesicles and exosomes released by cancer cells may play a significant role in the origin and remodeling of PMNs. They induce angiogenesis, alter permeability, and interact with proinflammatory cytokines. These vesicles may also promote the differentiation of normal breast fibroblasts into cancer-associated fibroblasts (CAFs) or prometastatic CAFs [109,110]. Within PMNs, an unstable virtual equilibrium governed by opposing immunological and nonimmunological, microenvironmental, or intratumor signals occurs, keeping cancer cells in a quiescent state. However, they can exit this quiescent state in response to various tumor microenvironmental cues, leading to cancer cell proliferation. Consequently, the Gompertz mathematical model of tumor growth, introduced by Goldie and Coldman, does not adequately capture the specific biology of disseminated cancer cells (DCCs) as described in the past two decades. In the context of locally advanced or high-risk breast cancer, we have proposed specific schedules of immunotherapy, cyclically administered in conjunction with or alternated with prolonged conventional adjuvant hormone therapy or a few cycles of chemotherapy. This proposed protocol [10,106] is cost-effective and straightforward, and we expect that several oncologic centers will initiate prospective randomized studies for its validation in the near future. Table 3 shows the main advantages and drawbacks of the targeted therapies in advanced breast cancer.

## 4. Discussion 

In the last two decades, several promising research avenues beyond targeted therapies have emerged for combating advanced breast cancer. In addition to drug repurposing, investigations focusing on breast cancer stem cells (CSCs), exosomes, T-lymphocyte infusions, and cancer cell reprogramming have garnered attention, with a particular emphasis on micrometastatic disease as a potential avenue for a definitive cure. However, these research directions have faced challenges in advancing and translating their findings into clinical trials. Interestingly, despite the clear limitations, the strategy of targeted therapies continues to receive significant support, especially in the context of common cancer types such as advanced breast cancer. Advances in basic research have unveiled the intricate nature of advanced breast cancer, and despite the many expectations, the benefits for patients from numerous clinical trials based on targeted therapies have been limited. Furthermore, the approved drugs that make it into clinical practice often come at a high cost covered by national healthcare services. The persistent focus on targeted-therapy-based strategies may have contributed to a reduced interest from pharmaceutical companies in sponsoring alternative research. Researchers dedicated to exploring different avenues have frequently encountered difficulties in securing collaboration. Recently, our research group and other scientists [10,106,111,112] following the advances in molecular biology and other experimental findings [113,114] have recognized micrometastatic disease as an ideal target for achieving a definitive cure for cancer. In response, we have outlined a feasible and innovative protocol for high-risk cancer patients, whether less or more biologically aggressive. This protocol involves the intermittent administration of conventional chemotherapy or endocrine therapy, alternated with novel immunotherapy schedules. We aspire for this cost-effective protocol to be swiftly implemented and evaluated in multicenter prospective randomized clinical trials. The goal is to significantly enhance the rate of high-risk cancer patients achieving definitive cures. However, while the success of these trials could represent a significant scientific achievement, it may not align with the interests of those who profit substantially from clinically overt metastatic disease. Consequently, it is not surprising that a majority of financial support continues to endorse a therapeutic strategy that primarily extends OS and/or PFS rather than actively targeting micrometastatic disease. While targeted therapies have contributed to the discovery of novel biologic prognostic and predictive biomarkers, along with new treatment options, it is essential to recognize that other treatments designed to counteract metastatic disease have often been restricted. 

## 5. Conclusions

In summary, this paper aims to stimulate a broader scientific discussion, encouraging the exploration of optimal approaches to address the mentioned limitations of targeted therapies in comparison with alternative therapeutic research avenues. It also underscores the potential of a definitive cure, as highlighted by recent advances in basic research.

## Figures and Tables

**Figure 1 cancers-16-00466-f001:**
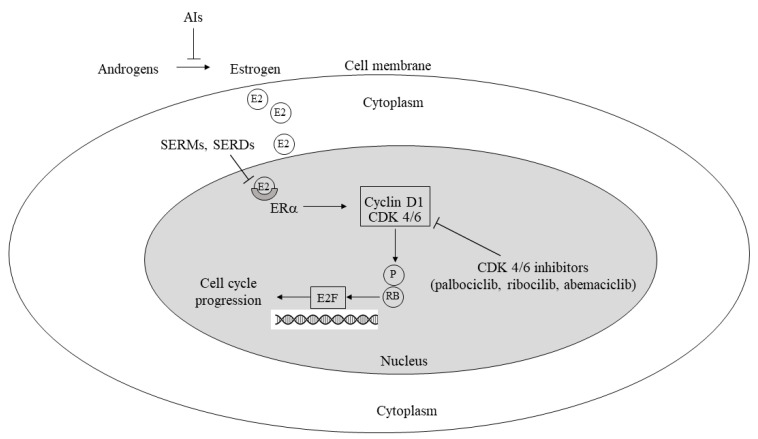
Mechanisms involved in cell cycle progression/inhibition by estrogens, antiestrogens, and CDK 4/6 inhibitors. AI: aromatase inhibitor; P: phosphorylation; SERM: selective estrogen receptor modulator; SERD: selective estrogen receptor degrader; CDK: cyclin-dependent kinase; RB: retinoblastoma protein; E2F: E2F transcription factor (also see text).

**Figure 2 cancers-16-00466-f002:**
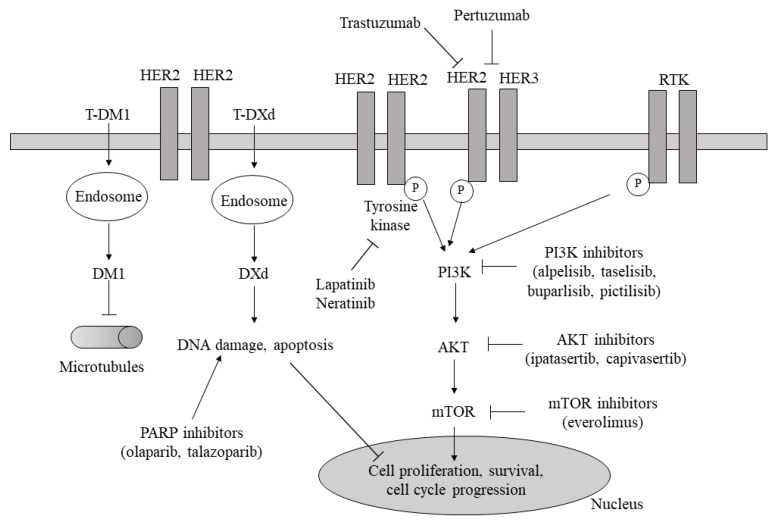
Signaling pathways involved in cell proliferation, survival, and cycle progression with the corresponding categories of inhibitors (anti-HER2 target therapies, PI3K/AKT/mTOR inhibitors, and PARP inhibitors). RTK: receptor tyrosine kinase; HER2/3: human epidermal growth factor receptor 2/3; P: phosphorylation; T-DM1: trastuzumab-emtansine; T-DXd: trastuzumab deruxtecan; PARP: poly-ADP ribose polymerase; PI3K: phosphoinositide 3-kinase; Akt: protein kinase; mTOR: mammalian target of rapamycin (also see text).

**Table 1 cancers-16-00466-t001:** Main characteristics and outcomes in clinical trials carried out with PIK3CA-Akt-mTOR or cyclin kinase inhibitors in advanced/metastatic breast cancer patients.

Main Characteristics											Outcome			Refs.
Clinical Trial	Molecular Subtype	Intervention	PTS (n)		Type of Drug		Study Arm		Control		PFS (mo)	OS (mo)	G3-4 AEs	
			Study Arm	Control	Study Arm	Control	LA/RC	M	LA/RC	M				
Bolero-2	HR+/HER2−	Eve + ae vs. ae	485	239	Exe	Exe	386	99	139	100	7.8 (11) vs. 3.2 (4.1)	31 vs. 26.6	42%	[13,14]
Lotus	TNBC	Ipatasertib + CT vs. CT	62	62	Pxt	Pxt	NA	NA	NA	NA	6.2 vs. 4.9	25.8 vs. 16.9	>15%	[20,21]
Pakt	TNBC	Capivesertib + CT vs. CT	70	70	Pxt	Pxt	0	70	0	70	5.9 vs. 4.2	19.1 vs. 12.6	>15%	[22]
Monaleesa-2	HR+/HER2−	Ribociclib plus ae vs. ae	334	334	Let (AI)	Let (AI)	1 (0.3%)	333 (99.7%)	3 (0.9%)	331 (99.1%)	25.3 vs. 16	63.9 vs. 51.4	>10%	[25,26]
Paloma-1	HR+/HER2−	Palbociclib plus ae vs. ae	444	222	Let (AI)	Let (AI)	3 (4%)	81 (96%)	1 (1%)	80 (99.4%)	27.6 vs. 14.5	53.9 vs. 51.2	>15%	[28,29]
Monarch-3	HR+/HER2−	Abemaciclib plus ae vs. ae	328	165	Let/Ana	Let/Ana	0	328 (100%)	0	165 (100%)	28.2 vs. 14.8	NA	58%	[27]

LA/RC: locally advanced/recurrent disease; PI3CA: phosphatidylinositol-4-5-biphosphate 3-Kinase catalytic subunit alpha; Akt: protein kinase b; mTOR: mammalian target of rapamycin; HR: hormone receptor; HER2: human epidermal growth factor 2; ae: antiestrogen; CT: chemotherapy; TNBC: triple-negative breast cancer; PTX: paclitaxel; Eve: everolimus; Exe: exemestane: M: metastatic; PFS: progression-free survival; OS: overall survival; AE: adverse event; AI: aromatase inhibitor; Let: letrozole; Ana: anastrozole; NA: not available.

**Table 2 cancers-16-00466-t002:** Main characteristics and findings in principal clinical trials carried out with anti-HER2 mAbs, ICIs, and PARP inhibitors as first-line therapy in advanced/metastatic breast cancer patients.

Main Characteristics										Outcome		G3-4 AEs (%)	Refs.
Clinical Trial	Molecular Subtype	Setting	Intervention	Pts (N)		STUDY ARM		Control					
				Study Arm	Control	LA/ LRI	M	LA/ LRI	M	mPFS (mo)	mOS (mo)		
Randomized phase III	HER2+	M	TRST + CT ^a^ vs. CT ^b^	235	234	0	235	0	234	7.4 vs. 4.6 (*p* < 0.001)	25.1 vs. 20.3 (*p* < 0.001)	27 or 13 ^a^ vs. 8 or 1 ^b^	[5]
Cleopatra	HER2+	M	TRST + PRTZ + DTX vs. TRST + DTX + PBO	402	406	0	402	0	406	18.5 vs. 12.4 (*p* < 0.001)	57.1 vs. 40.8 (*p* < 0.001)	Similar in the 2 groups	[36,37]
Keynote-355	TNBC	A	PE + CT ^c^ vs. PBO + CT ^c^	566	281	383/13	167	185/12	84	9.7 vs. 5.6 (*p* = 0.0012) 7.6 vs. 5.6 (*p* = 0.0014) 7.5 vs. 5.6 (*p* n.s.) (ITT)	23 vs. 16.1 (*p* = 0.0019) (CPS > 10) 17.6 vs. 16 (*p* n.s.) (CPS > 1) 17.2 vs. 15.5 (*p* n.s.) (ITT)	68.1 vs. 66.9	[63,64]
Impassion-130	TNBC	A	ATZ + Nab-PTX vs. Nab-PTX + PBO	451	451	47	404	43	408	7.2 vs. 5.5 (*p* = 0.002) (ITT) 7.5 vs. 5.5 (*p* < 0.001) (PDL1+)	21 vs. 18.7 (*p* n.s.) (ITT) 25.4 vs. 17.9 (*p* n.s.) (PDL1+)	16.7 vs. 12.9	[65,66]
Impassion-131	TNBC	A	ATZ + PTX vs. PTX + PBO	191	101	135	56	71	30	6 vs. 5.7 (*p* n.s.) (PDL1+)	22.1 vs. 28.3 (*p* n.s.) (PDL1+)	11 vs. 5	[67]
OlympiA-D	gBRCA mutation, HER2−	M	OLP vs. CT ^d^	205	97	0	205	0	97	7 vs. 4.2 (*p* < 0.001)	19.3 vs. 17.1 (*p* n.s.)	36.6. vs. 50.5	[68,69]
EMBRACA	gBRCA mutation, HER2−	A	TLZ vs. CT ^e^	287	144	15	271	9	135	8 vs. 5.6 (*p* < 0.001)	19.3 vs. 19.5 (*p* n.s.)	55 vs. 38 (H) 32 vs. 28 (NH)	[70,71]

HER2: human epidermal growth factor receptor 2; mAbs: monoclonal antibodies; gBRCA: germline BRCA; ICIs: immune checkpoint inhibitors; PARP: poly ADP ribose polymerase; TNBC: triple-negative breast cancer; TRST: trastuzumab; CT: chemotherapy; LA: locally advanced; LRI: locally recurrent inoperable; M: metastatic; A: advanced; mPFS: mean progression-free survival; mOS: mean overall survival; AE: adverse event; DTX: docetaxel; PRTZ: pertuzumab; PBO: placebo; PE: pembrolizumab; ATZ: atezolizumab; OLP: olaparib; TLZ: talazoparib; H: hematologic; NH: nonhematologic. ^a^ doxorubicin (DOX) and cyclophosphamide (CY) (143) or paclitaxel (PTX) (92); ^b^ DOX (epirubicin in 36) and CY (138) or PTX (96); ^c^ nabPTX or PTX or gemcitabine (GEM) + carboplatin (CBDCA); ^d^ capecitabine, eribulin, or vinorelbine; ^e^ capecitabine, eribulin, GEM, or vinorelbine.

**Table 3 cancers-16-00466-t003:** Main advantages and drawbacks of the targeted therapies in advanced breast cancer.

Usefulness	Drawbacks
Basic research	(1) High cost/benefit ratio (2) Arising of resistance, as for conventional therapy (3) AE increase for combined therapy and poor QoL (4) Great restraint of other different lines of research
Elucidation of: (1) Molecular pathways (2) Cell-to-cell signaling and interactions in TME (3) Genetic and epigenetic alterations (4) Prognostic and predictive biomarkers	
Clinical practice	
(5) Some specific therapeutic options with improved outcomes in selected populations	

TME: tumor microenvironment; AEs: adverse events.

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
