# Peer review of "Targeted Therapies and Drug Resistance in Advanced Breast Cancer, Alternative Strategies and the Way beyond"

_cancers, 2024, doi:10.3390/cancers16020466_

Round 1
Reviewer 1 Report
Comments and Suggestions for Authors
This review article is focused on therapies, including targeted therapies, for women with breast cancer. The review goes in depth into the different therapies used to treat women with different stage disease and some of the known mechanisms of drug resistance. In addition, the authors address the idea that perhaps too much emphasis has been given to the precision medicine approach over the past few decades and they make the case that other treatments/approaches should not be abandoned. In particular, the authors suggest that drug repurposing and a focus on treatments for patients with micrometastasis are important avenues that warrant investigation. Overall, this is a great in depth review of the literature and this reviewer finds the author’s arguments compelling that an overemphasis on a precision medicine approach is not in the best interest of all patients with breast cancer or progress in the field.
There are no major concerns with the manuscript, although editing is warranted. A list of some examples are below.
One overarching change that should be made throughout the manuscript is how patients with breast cancer is discussed/written. For example, it is proper to refer to patients with breast cancer rather than breast cancer patients. Or, for instance, patients with metastatic breast cancer, rather than metastatic breast cancer patients.
abstract.
1. “During this prolonged time, research….”. Should be changed to “During this time, research…”
2. “Commonly, a minority of them are successful…” should be changed to “Commonly, a minority of them have been successful…”
3. “The diffusion of this expensive therapeutic strategy…” Diffusion is perhaps not the best word choice here.
4. “…although they often did not find the required collaboration.” Perhaps something along the line of “…although they are often not pursued due to a focus on precision medicine “
5. “…and the way beyond of research…” needs to be rewritten.
Introduction
1. ER+/HER2- luminal breast cancer comprehends…” Perhaps “comprises” rather than “comprehends”
2. “…focused on molecular pathways fostering the main cancer hallmarks.” This needs to be rewritten.
3. The last sentence of the introduction also needs to be rewritten. Would be better to simplify the sentence…issues with this therapeutic strategy that are examined…
section 2.
1. rewrite the first sentence.
2. The term “The used definition….” does not make sense.
section 2.1
1. line 4. check spelling.
2. end of paragraph. Fig 1-2 should be Figs 1-2.
section 2.1.1.
1. The first sentence should be rewritten “…of an altered signaling…”?
2. “Accordingly, in aggressive…. was considered an helpful option of …” perhaps “Accordingly, in aggressive…. was considered a helpful option for …”
3. “…the former is on evaluation in a Phase…” perhaps “…the former is undergoing evaluation in a Phase…”
Perhaps the paragraph starting with “Monoclonal antibodies (mAb) against Her2…” should be section 2.1.2?
same section
1. “HER2+ breast cancer subtype has worse prognosis…” Worse prognosis than?
2. “Nevertheless, due to intrinsic or arising of resistance…” arising of resistance?
3. “…trastuzumab via a non-reducible thioeter…” should this be “…trastuzumab via a non-reducible thioether…”
4. “The efficacy of T-DX-d which comprehends…” perhaps “The efficacy of T-DX-d which is comprised of…”
5. “Particularly, CTLA-4 signaling, unlike CD28 signals which are necessary…” perhaps “Particularly, CTLA-4 signaling, unlike CD28 signaling is necessary…”
6. “Therefore, ICB by ICIs… strategy aiming to remove…” perhaps “Therefore, ICB by ICIs… strategy to remove…”. The second part of this sentence also needs to be rewritten.
7. Approximately, 10%-20% of TNBC patients display gBRCA mutations…” BRCA1/2 germline (gBRCA) should be written out the first time the abbreviation is used.
Comments on the Quality of English Languagesee comments above
Author Response
Answers to remarks
Reviewer 1
Abstract
Question 1. “During this prolonged time, research…” Should be changed to “During this time, research…”
Answer: The suggested change has been made.
Question 2. “Commonly, a minority of them are successful…” should be changed to “Commonly, a minority of them have been successful…”
Answer: As suggested by language editing, the sentence become “Commonly, only a minority of them are successful…”.
Question 3. “The diffusion of this expensive therapeutic strategy…” Diffusion is perhaps not the best word choice here.
Answer: As suggested by language editing, the word “diffusion” has been replaced with “spread”.
Question 4. “…although they often did not find the required collaboration.” Perhaps something along the line of…although they are often not pursued due to a focus on precision medicine”
Answer: The suggested change has been made.
Question 5. “…and the way beyond of research…” needs to be rewritten.
Answer: As suggested by language editing “…and the way beyond of research…” has been re-written as follows “a way beyond the current research”.
Introduction
Question 1. ER+/HER2- luminal breast cancer comprehends…” Perhaps “comprises” rather than “comprehends”
Answer: As suggested by language editing the sentence became “ER+/HER2- luminal breast cancer accounts for…”
Question 2. “…focused on molecular pathways fostering the main cancer hallmarks”. This needs to be rewritten.
Answer: As suggested by language editing this sentence has been rewritten as follows “focused on molecular signalling that promote the main cancer hallmarks.”
Question 3. The last sentence of the introduction also needs to be re-written. Would be better to simplify the sentence…issues with this therapeutic strategy that are examined…
Answer: As suggested by language editing, this sentence has been rewritten as follows “this paper examines some principal issues with this therapeutic strategy in advanced breast cancer, and provides insights into the way forward”.
Section 2.
Question 1. rewrite the first sentence.
Answer: As suggested by language editing, the first sentence “Precision medicine or targeted therapies widespread with the backing of multinational drug companies” has been replaced with “Precision medicine, heavily supported by multinational drug companies……”
Question 2. The term “The used definition….” does not make sense
Answer: The term “The used definition….” has been removed.
Section 2.1
Question 1. Line 4. check spelling.
Answer: line 4. check spelling has been made and as suggested by language editing the sentence “the initial setting fo ther investigation” has been replaced with “Advanced disease often serves as the initial setting for therapeutic investigation”
Question 2. End of paragraph. Fig 1-2 should be Figs 1-2.
Answer: “Fig 1-2” has been replaced with “Figs 1-2.”
Section 2.1.1.
Question 1. The first sentence should be rewritten “…of an altered signaling…”?
Answer: The first sentence “Involvement in endocrine resistance of an altered signalling through the PI3K/AKT/mTORC1 molecular pathway suggested the association of the PI3K/AKT/mTORC1 pathway inhibitors with conventional endocrine therapy” has been rewritten and replaced with “Involvement in endocrine resistance of an altered PI3K/AKT/mTORC1 signalling suggested the association of the PI3K/AKT/mTORC1 inhibitors with conventional endocrine therapy”
Question 2. “Accordingly, in aggressive…. was considered an helpful option of …” perhaps “Accordingly, in aggressive…. was considered a helpful option for …”
Answer: As suggested by language editing, the sentence has been replaced with “is considered a helpful treatment option”
Question 3. “…the former is on evaluation in a Phase…” perhaps “…the former is undergoing evaluation in a Phase…”
Answer: As suggested the sentence has been replaced with “the former is undergoing evaluation in a Phase II clinical trial”
Question. Perhaps the paragraph starting with “Monoclonal antibodies (mAb) against Her2…” should be section 2.1.2?
Answer: the paragraph starting with “Monoclonal antibodies (mAb) against Her2…” does not take part of be section 2.1.2. In fact, this sentence became the title of section 2.1.3 and the following section became section 2.1.4.
Same section
Question 1. “HER2+ breast cancer subtype has worse prognosis…” Worse prognosis than?
Answer: The sentence has been completed as follows “The HER2+ subtype of breast cancer, known for its poorer prognosis compared to luminal breast cancer”
Question 2. “Nevertheless, due to intrinsic or arising of resistance” arising of resistance?
Answer: A definition of the terms used has been provided (also see reviewer 2) and, when the sentence has been first mentioned, it has been replaced with “constitutive (intrinsic or “de novo”) or acquired drug resistance as result of initial genetic aberrations or arisen during disease progression respectively”
Question 3. “…trastuzumab via a non-reducible thioeter…” should this be “…trastuzumab via a non-reducible thioether…”
Answer: As suggested the sentence has been replaced with “via a non-reducible thioether link”
Question 4. “The efficacy of T-DX-d which comprehends…” perhaps “The efficacy of T-DX-d which is comprised of…”
Answer: As suggested the sentence has been replaced with “The efficacy of T-DXd which is comprised of an anti-HER2 antibody”
Section 2.1.4
Question 5. “Particularly, CTLA-4 signaling, unlike CD28 signals which are necessary…” perhaps “Particularly, CTLA-4 signaling, unlike CD28 signaling is necessary…”
Answer: CD-28 and CTLA-4 play opposite action on T lymphocytes. In fact, CD-28 stimulates while CTLA-4 signaling inhibits T lymphocytes activity. Therefore, in the following paragraph “Indeed, cluster of differentiation 28 (CD28) and cytotoxic T-lymphocyte associated antigen 4 (CTLA-4) are other co-stimulatory signals which play a principal role in regulating T-cell activity. Particularly, CTLA-4 signalling unlike CD28 signals, which are necessary to activate T-cells and for cytokine release, counteracts T-cell activation.” The adjective “co-stimulatory” has been removed and the paragraph as suggested by language editing became as follows “Signals such as cluster of differentiation 28 (CD28) and cytotoxic T lymphocyte associated antigen 4 (CTLA-4) play a pivotal role in regulating T-cell activity; CTLA-4 signaling, in contrast to CD28, which is necessary to activate T cells and for cytokine release, inhibits T-cell activation”
Question 6. “Therefore, ICB by ICIs… strategy aiming to remove…” perhaps “Therefore, ICB by ICIs… strategy to remove…”. The second part of this sentence also needs to be rewritten.
Answer: The sentence “Therefore, ICB by ICIs has become a novel targeted therapeutic strategy aiming to remove the ‘brake’ to anti-tumour immune response by hindering these molecules inhibiting effector T-cells” as suggested by language editing has been replaced with “Therefore, ICB by ICIs has emerged as a novel strategy to lift the 'brake' on the anti-tumor immune response”.
Question 7. Approximately, 10%-20% of TNBC patients display gBRCA mutations…” BRCA1/2 germline (gBRCA) should be written out the first time the abbreviation is used.
Answer: The sentence “Approximately 10%-20% of TNBC patients display gBRCA mutation and in these BRCA1/2 germline (gBRCA) mutation carriers, PARPi are advised.” Has been replaced with “Approximately 10%-20% of TNBC patients exhibit a BRCA1/2 germline (gBRCA1/2) mutation, and PARP inhibitors (PARPi) are recommended for these individuals.”

Reviewer 2 Report
Comments and Suggestions for Authors
This review article by Nicolini and Ferrari describes targeted therapeutic protocols in the case of breast cancer, from a critical perspective, as well as drug resistance mechanisms and unconventional/alternative strategies to address current challenges in targeting of the disease. The review is quite inclusive of recent and relevant literature, as well as clinical studies and their results in this field, providing an informed and authoritative coverage of the topic.
I recommend publication after the following minor revisions:
(1) Figure 1 is somewhat misleading. Please provide an amended figure with a clearer representation of cellular compartments and membrane-related events. Also enlarge font to make more legible.
(2) Figure 2: Enlarge font and color-code the different signaling pathways.
(3) In p. 6 there's a sentence break-up at the beginning of the page.
(4) Please provide definition for terms used:
p. 10, paragraph 1: "constitutive" vs. "arising from drugs resistance"
p. 13, paragraph 2: "de novo or acquired drug resistance"
(5) Paragraph 2.4 ("The main limitations") needs to be developed more, by inclusion of appropriate examples.
(6) Inclusion of an abbreviations list is recommended.
Comments on the Quality of English Language
Minor improvements in the use of English are needed throughout the text.
Author Response
Answers to remarks
Reviewer 2
Question (1) Figure 1 is somewhat misleading. Please provide an amended figure with a clearer representation of cellular compartments and membrane-related events. Also enlarge font to make more legible.
Answer: Figure 1 has been accordingly amended.
Question (2) Figure 2: Enlarge font and color-code the different signaling pathways.
Answer: In Figure 2 font has been enlarged; perhaps a color-code for the different signaling pathways is no more necessary.
Question (3) In p. 6 there is a sentence break-up at the beginning of the page.
Answer: The comment is right, this sentence became the title of section 2.1.3. Therefore, the following section became section 2.1.4. (Also see reviewer 1); see page 6, lines 1-2.
Question (4) Please provide definition for terms used:
Section 2.1.3. p. 10, paragraph 1: "constitutive" vs. "arising from drugs resistance"
Section 2.2, p. 13, paragraph 2: "de novo or acquired drug resistance"
Answer: Definition for "constitutive" vs. "arising from drugs resistance/acquired drug resistance “
has been provided when first mentioned", see page 6, lines 18-19 and page 10, line 2.
Question (5) Paragraph/Section 2.4 ("The main limitations") needs to be developed more, by inclusion of appropriate examples.
Answer: "The main limitations" paragraph has been developed and two examples have been added, see page 10, lines 39-49.
Question (6) Inclusion of an abbreviations list is recommended.
Answer: An abbreviation list has been added at the end of manuscript.
Reviewer 3 Report
Comments and Suggestions for Authors
The manuscript by Nicolini et al, entitled “Targeted therapies and drugs resistance in advanced breast cancer, alternative strategies and the way beyond” provides an overview of the recent clinical trials in breast cancer. However, I have the following concerns as follows:
Comments
1. All abbreviations used in Figures and Tables should be defined in the legend.
2. Most references are review papers and the original papers referenced in the manuscript are too old (more than 5 years). Therefore, the manuscript lacks novelty.
3. In the section 2.2.1. This subsection is the summary of the author’s previous review. This is not appropriate for this review paper.
4. Section 3. This section might be included in the Discussion section because there appears to be a lack of material.
Comments on the Quality of English LanguageThe manuscript would benefit tremendously from language editing by either a native English speaker or a professional editor. Particularly in section 2. Also, there are a number of typographical errors throughout the manuscript.
Author Response
Answers to remarks
Reviewer 3
Question 1. All abbreviations used in Figures and Tables should be defined in the legend
Answer: All abbreviations used in Figures and Tables have been defined in the legend and added at the end of the manuscript.
Question 2. Most references are review papers and the original papers referenced in the manuscript are too old (more than 5 years). Therefore, the manuscript lacks novelty.
Answer: Many references are research papers related to principal clinical trials that were carried out and shown in tables 1-2. Some of them are historical clinical trials and papers report first the pre-established final endpoint/s of the clinical trial. Due to this, a few of them are “more than 5 years old”. As to the cited reviews, they are those strictly functional/related to the paper content.
Question 3. In the section 2.2.1. This subsection is the summary of the author’s previous review. This is not appropriate for this review paper.
Answer: Subsection 2.2.1 has been strongly shortened and for more details readers have been addressed to the reported review article.
Question 4. Section 3: This section might be included in the Discussion section because there appears to be a lack of material.
Answer: As suggested section 3 has been included in the Discussion.
Round 2
Reviewer 3 Report
Comments and Suggestions for Authors
The manuscript has been much improved. I have just a few minor comments, which I believe will improve the readability of the paper.
1. P6, Line 175-177. “The authors concluded that those who were given T-DXd were more likely to have disease progression or death less than those who were given TD-M1 [49].”
Is the description correct?
2. P7, Line 202-204. “PD-1, CTLA-4 and CD28 which are receptors on 202 effector T cells bind with their ligands including CD274 (PD-L1) and CD273 (PD-L2) among others [58–59]”
The sentence is confusing because the description of the ligand for CTLA-4 and CD28 are lacking.
3. Section 2.1.4. To make it easier for readers to read, please change paragraphs when changing the topics.
4. Table 2. In Cleopatra clinical trial. Please describe the p-value instead of ??? in mOS.
5. There are still several typos in this paper. The authors should proofread their paper carefully.
Author Response
Answers to remarks
Reviewer 1
The manuscript has been much improved. I have just a few minor comments, which I believe will improve the readability of the paper.
Comment 1
P 6, Line 175-177.The authors concluded that those who were given T-DXd were more likely to have disease progression or death less than those who were given TD-M1 [49].” Is the description correct?
Answer
The description is correct however to improve the clarity of the sentence it has been replaced as follows “The authors concluded that those who were given T-DXd were more likely to have lower risk of disease progression or death than those who were given TD-M1 [49].”
Comment 2
P7, Line 202-204. “PD-1, CTLA-4 and CD28 which are receptors on effector T cells bind with their ligands including CD274 (PD-L1) and CD273 (PD-L2) among others [58–59]”
The sentence is confusing because the description of the ligand for CTLA-4 and CD28 are lacking.
Answer
P7, Line 202-204. To better clarify the description this sentence has been replaced with “PD-1, CTLA-4 and CD28 are receptors on effector T cells and they bind with ligands which are CD274 (PD-L1) and CD273 (PD-L2) for PD-1 and B7-1 (CD80) or B7-2 (CD86) among others for CTLA-4 and CD28 [58–59]”
Comment 3
Section 2.1.4. To make it easier for readers to read, please change paragraphs when changing the topics.
Answer
Section 2.1.4. As suggested, to make easier for readers to read, when changing the topics we got to the point.
Comment 4
Table 2. In Cleopatra clinical trial. Please describe the p-value instead of ??? in mOS.
Answer
Table 2, In Cleopatra clinical trial p<0.001 has been added as requested
Comment 5
There are still several typos in this paper. The authors should proofread their paper carefully.
Answer
The paper has been carefully proofread and the typos have been accordingly removed.
